# Role of cognitive ability in the association between functional health literacy and mortality in the Lothian Birth Cohort 1936: a prospective cohort study

Chloe Fawns-Ritchie,[1,2] John M Starr,[1,3] Ian J Deary[1,2]

[1]Centre for Cognitive Ageing and Cognitive Epidemiology, University of Edinburgh, Edinburgh, UK
[2]Department of Psychology, University of Edinburgh, Edinburgh, UK
[3]Alzheimer Scotland Dementia Research Centre, University of Edinburgh, Edinburgh, UK

**Correspondence to**
Professor Ian J Deary;
i.deary@ed.ac.uk

## ABSTRACT

**Objectives** We investigated the role that childhood and old age cognitive ability play in the association between functional health literacy and mortality.

**Design** Prospective cohort study.

**Setting** This study used data from the Lothian Birth Cohort 1936 (LBC1936) study, which recruited participants living in the Lothian region of Scotland when aged 70 years, most of whom had completed an intelligence test at age 11 years.

**Participants** 795 members of the LBC1936 with scores on tests of functional health literacy and cognitive ability in childhood and older adulthood.

**Primary and secondary outcome measures** Participants were followed up for 8 years to determine mortality. Time to death in days was used as the primary outcome measure.

**Results** Using Cox regression, higher functional health literacy was associated with lower risk of mortality adjusting for age and sex, using the Shortened Test of Functional Health Literacy in Adults (HR 0.95, 95% CI 0.92 to 0.98), the Newest Vital Sign (HR 0.88, 95% CI 0.80 to 0.97) and a functional health literacy composite measure (HR 0.77, 95% CI 0.65 to 0.92), but not the Rapid Estimate of Adult Literacy in Medicine (HR 0.95, 95% CI 0.90 to 1.01). Adjusting for childhood intelligence did not change these associations. When additionally adjusting for fluid-type cognitive ability in older age, associations between functional health literacy and mortality were attenuated and non-significant.

**Conclusions** Current fluid ability, but not childhood intelligence, attenuated the association between functional health literacy and mortality. Functional health literacy measures may, in part, assess fluid-type cognitive abilities, and this may account for the association between functional health literacy and mortality.

## INTRODUCTION

Health literacy is "the degree to which individuals have the capacity to obtain, process and understand basic health information and services needed to make basic health decisions".[1] This ability is thought to be

### Strengths and limitations of this study

► This study used three functional health literacy tests, which enabled us to create a composite functional health literacy measure.
► This study had comprehensive tests of cognitive ability measured in both childhood and old age which allowed us to investigate whether childhood and old age cognitive ability independently played a role in the relationship between functional health literacy and mortality.
► The health literacy measures used here only assessed functional health literacy, and therefore, we cannot determine whether cognitive ability would attenuate the association between health literacy and mortality if we used multidimensional health literacy measures.
► Larger samples and a longer follow-up time are needed to determine the role of cognitive ability in the association between functional health literacy and cause-specific mortality.

multifaceted and encompasses the set of skills required to navigate the healthcare environment.[2-4] One component of health literacy is functional health literacy—the reading, writing and numeracy skills required to understand health information.[3 5 6] Tests designed to assess functional health literacy have been developed to measure health-related reading and numeracy skills, such as the commonly used Test of Functional Health Literacy in Adults.[5 6] This test requires participants to read materials often used in the healthcare setting, such as a medicine bottle, and answer questions about these materials.

Performance on functional health literacy tests has been associated with a range of health outcomes. Individuals with lower functional health literacy are more likely to require emergency care and have poorer skills in relation to correctly taking medication and

**BMJ**

interpreting written health materials.[7] Individuals with higher functional health literacy are more likely to take part in health-promoting behaviours, such as eating a healthy diet, and are more likely to take part in routine cancer screening.[8 9]

Successful completion of functional health literacy measures relies on cognitive functions, such as processing capacity and reasoning.[10 11] One dominant theory in intelligence research is that there is a distinction between fluid ability, the ability to problem solve using novel material, which tends to decline with increasing age, and crystallised ability, which is the knowledge acquired throughout life which remains relatively stable across the lifespan.[12–16] Successful completion of tests of functional health literacy likely requires both crystallised abilities, such as specific knowledge relating to health, and fluid abilities, such as reasoning.[10 11] It is therefore unsurprising that performance on tests of functional health literacy and cognitive function are strongly related.[17–24] Some tests of functional health literacy have been found to correlate more strongly with measures of cognitive ability than with each other.[23 25 26] This overlap is so strong that some have proposed that functional health literacy should not be considered a unique construct but, instead, should be thought of as a specific component of cognitive function.[26]

Given the association between performance on tests of functional health literacy and cognitive ability tests, researchers have investigated whether the relationship between functional health literacy and health remains when also measuring cognitive ability. Whereas most evidence suggests that cognitive function explains a large proportion of the association between functional health literacy and health, the degree of attenuation varies.[25 27 28] A study using participants from the Lothian Birth Cohort 1936 (LBC1936)[25]—the same sample used in the current study—investigated whether cognitive ability in childhood and late adulthood attenuated the association between functional health literacy and physical health. In models without cognitive function, functional health literacy was associated with all three of the measures of physical health assessed. Addition of cognitive ability in older age significantly attenuated the association between functional health literacy with physical fitness by 43% and number of natural teeth by 39%; however, it did not attenuate the association between functional health literacy and body mass index (BMI). Conversely, whereas childhood cognitive ability did not attenuate the association between functional health literacy and physical fitness, it attenuated the association between functional health literacy and number of teeth by 30% and BMI by 88%. In the fully adjusted model which included childhood and late adulthood cognitive ability, as well as other early-life factors, the association between functional health literacy and physical fitness, though attenuated by 43%, remained significant,[25] suggesting that functional health literacy may play a small but unique role in physical fitness.

Mortality is arguably one of the most important health outcomes to examine. Both cognitive ability[29 30] and functional health literacy[31] have been found to predict mortality. Researchers have therefore investigated the degree to which cognitive function explains the association between functional health literacy and mortality. When not controlling for cognitive function, Baker *et al*[32] found that individuals with inadequate compared with adequate health literacy had a 50% higher risk of dying. When additionally adjusting for cognitive function, the risk reduced to 27%, but remained significant. A similar pattern of attenuation was found in another study.[33] Thus, cognitive function did not fully explain this relationship. These two studies, however, used brief measures of functional health literacy and cognitive function.

The present study sought to better understand the relationship between functional health literacy, cognitive ability and mortality using data from the LBC1936. We note that this is the same sample as used in Mõttus *et al*[25] to investigate the association between functional health literacy, cognitive ability and physical health. In this previous study,[25] physical health was measured concurrently with fluid ability and functional health literacy. The current analysis is different from and complementary to this previous study in that we followed up the participants for 8 years to determine mortality status—obviously a most important health outcome. Studies that have examined the role that cognitive function plays in the association between functional health literacy and mortality used brief cognitive measures collected at the same time as the functional health literacy tests.[32 33] It is not known whether early-life cognitive ability and cognitive ability in older age play different roles in the association between health literacy and mortality. The current analysis utilises cognitive test scores collected in childhood, which are thought to measure the trait of lifelong intelligence, and current cognitive ability in older age, measured at approximately 73 years and contemporaneously with functional health literacy. The aim of this study was to determine whether childhood cognitive ability and current cognitive ability in older adulthood play unique roles in the association between functional health literacy and mortality.

## METHODS

### Participants

LBC1936 is a cohort study of 1091 older adults born in 1936, most of whom reside in the Lothian area in Scotland. Most had taken part in the Scottish Mental Survey 1947, which tested the intelligence of almost all children born in 1936 and attending Scottish schools on 4 June 1947.[34] LBC1936 consists of a sample of these individuals who were subsequently followed up, for the first time, at age 70 years (wave 1). To date, these participants have been followed up a further three times at approximately 3-year intervals (waves 2–4). LBC1936 was designed principally to investigate healthy, non-pathological, cognitive ageing. Detailed information on this cohort is provided

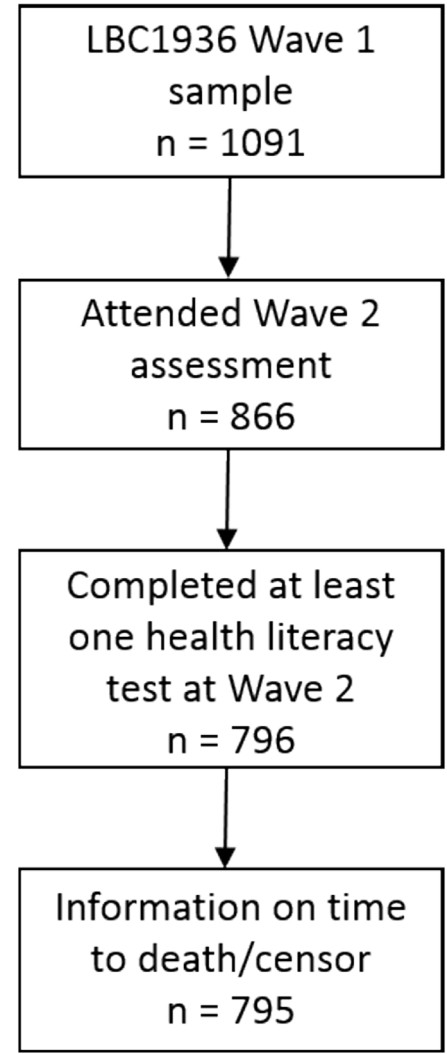

**Figure 1** Flow diagram of the sample used to investigate the role of cognitive ability in the association between health literacy and mortality (n=795). LBC1936, Lothian Birth Cohort 1936.

elsewhere.[35][36] The present study used a subsample of 795 (413 male and 382 female) LBC1936 participants who completed tests of health literacy at wave 2 when participants were approximately aged 73 years. Figure 1 shows a flow chart of how the analytic sample for this current study was derived.

Written informed consent was obtained from participants. This study conformed to the principles embodied in the Declaration of Helsinki.

## Measures

### Mortality and survival time

The General Register Office for Scotland was used to identify deaths. Deaths through the end of March 2017 were recorded, and this date is used as the censoring date for participants who survived. Survival time was measured in days from date of attending study visit at wave 2 to date of death or censoring date.

### Functional health literacy

Three functional health literacy tests were administered at wave 2.

*Rapid Estimate of Adult Literacy in Medicine (REALM)*[37]: This test measures participants' ability to read and correctly pronounce medical words. Participants are presented a piece of paper with a list of 66 medical words and are asked to read these words aloud. The words range in difficulty from easy ('fat') to difficult ('impetigo'). One point is given for each correctly pronounced word. One week test–retest ($r$=0.99)[37] and internal consistency (Cronbach's alpha=0.98)[38] have been found to be very high.

*Shortened Test of Functional Health Literacy in Adults (S-TOFHLA)*[5][6]: In the numeracy section, participants are provided with cards with medical information on them and are asked four questions about this information. The reading comprehension section comprised a 36-item task which involved participants reading two health-related passages where every fifth to seventh word was missing, and participants were to select the missing word from four options. Participants had 12 min to complete both sections. Here, the British version of the S-TOFHLA[9] was used which substitutes the Medicaid passage for a passage about UK prescription fee exemptions. This measure is a shortened version of the Test of Functional Health Literacy in Adults, which is seen as the gold standard functional health literacy test.[39] Successful completion of the S-TOFHLA requires the ability to read and comprehend written words and numbers in a health context. Internal consistency is high for reading comprehension (Cronbach's alpha=0.97)[6] and adequate for numeracy (Cronbach's alpha=0.68).[6] The S-TOFHLA has been found to correlate strongly with the REALM ($r$=0.80).[6]

*Newest Vital Sign (NVS)*[40]: Participants were presented with a nutrition label from a container of ice cream and were asked to answer six questions about the information provided on this label. The NVS assesses both reading comprehension and numeracy skills associated with health as participants need to use the written text and numbers on the label to answer the questions.[40] The NVS correlates with the S-TOFHLA at $r$=0.59[40] and shows reasonable internal consistency (Cronbach's alpha=0.76).[40]

*General health literacy:* The three functional health literacy measures used here have been found to correlate moderately with each other.[25] To capture the shared variance between these tests, a general measure of functional health literacy was created by entering scores on the three tests into a principal component analysis (PCA). Two of these measures had skewed distributions (see online supplementary figures 1–8); therefore, Spearman's rank correlation was used in the PCA. Only the first component had an eigenvalue >1, and the scree slope indicated a single component; therefore, scores from the first unrotated principal component were used as a composite of functional health literacy (general functional health literacy). This component accounted for 59.7% of the

total variance, confirming that there was substantial shared variance between the three functional health literacy tests. The REALM, S-TOFHLA and NVS loaded 0.74, 0.80 and 0.77, respectively, on this component.

## Cognitive ability

### Childhood cognitive ability (age-11 IQ)

As part of the Scottish Mental Survey 1947, almost all 11-year-old children in Scotland in 1947 sat the Moray House Test No. 12 (MHT),[34] a 45-minute, group-administered intelligence test that included tasks of verbal reasoning and spatial ability and had a maximum score of 76. In LBC1936, scores on the MHT were adjusted for age in days at testing and then were converted into standard IQ-type scores with a mean of 100 and an SD of 15. This score will be used as a measure of prior, or crystallised, ability.

### Current fluid ability

Participants completed a lengthy cognitive assessment.[35 36] As has been done in previous LBC1936 studies,[23 25] six tests administered at wave 2 thought to measure fluid-type cognitive abilities that tend to decline across the lifespan[14–16] were entered into a PCA. The following tests from the Wechsler Adult Intelligence Scale—III[41] that assess non-verbal reasoning, visuospatial ability, working memory and processing speed were used: Matrix Reasoning, Block Design, Letter–Number Sequencing, Symbol Search, Digit Span Backwards and Digit Symbol-Coding. Only the first component had an eigenvalue >1, and the scree slope indicated one component, and therefore, scores from this first principal component were used as a measure of current fluid ability. This component accounted 50.2% of the total variance. The loadings for the six tests were: Matrix Reasoning=0.69, Block Design=0.71, Letter–Number Sequencing=0.71, Symbol Search=0.75, Digit Span Backwards=0.64 and Digit Symbol-Coding=0.75.

## Covariates

Sociodemographic variables included in this analysis were education and occupational social class. Years of full-time education completed, recorded at wave 1 when participants were aged 70 years, was used to measure education. At wave 1, participants were assigned to one of the following occupational social classes based on their highest occupational status prior to retirement[42]: professional, managerial and technical, skilled, partly skilled manual and unskilled manual. Female participants were assigned the occupational class of their husband if this was higher than their own. Skilled was separated into skilled non-manual and skilled manual. Only five participants in this sample were assigned the occupational class of unskilled; therefore, partly skilled manual and unskilled manual were combined into one class, hereafter referred to as manual (n=31).

Three measures of health status measured at wave 2 were used. Self-reported health was measured by asking participants whether they rated their general health to be excellent, very good, good, fair or poor. Only a small number of participants who were recorded dead at the censoring date reported poor (n=3) or excellent (n=17) health. Therefore, poor and fair were collapsed into one category (fair/poor; n=73), as were very good and excellent (very good/excellent; n=487). Total score on the Hospital Anxiety and Depression Scale (HADS)[43] was used as a measure of mood state. Higher scores on the HADS represent higher levels of anxiety and depression. Activities of daily living were assessed using the Townsend Disability Scale.[44] Participants were given a score of 0 (no difficulty completing this activity) to 2 (not able to complete this activity) for nine activities, and thus higher scores represent more functional disability.

## Patient and public involvement

LBC1936 participants were not involved in the development of any part of this study. The results will be disseminated to participants via a quarterly newsletter sent to LBC1936 participants.

## Statistical analysis

SPSS V.21.0 was used to carry out this analysis. To determine whether those recorded as alive or dead at censoring date differ on demographic, functional health literacy, cognitive function or health status variables, $X^2$ tests were conducted for categorical variables, independent t-tests were used for normally distributed continuous variables and Mann-Whitney U tests were used for non-normal continuous variables. Spearman's rank-order correlation was used to examine the relationship between functional health literacy and cognitive ability scores. To investigate the association between functional health literacy and time to death, Cox proportional hazard regression was used. For each of the functional health literacy measures of interest (REALM, S-TOFHLA, NVS and the composite score of general functional health literacy), six models were run. In Model 1, the functional health literacy measure of interest and age and sex was entered. Years of education was added in Model 2 as this has been found to be associated with functional health literacy. To determine whether cognitive ability in childhood attenuated the association between functional health literacy and mortality, age-11 IQ was added (Model 3). In Model 4, fluid-type cognitive ability in older age was additionally added to determine its role in the association between functional health literacy and mortality. Occupational class was additionally included in Model 5. Finally, health status variables (self-reported health, HADS and Townsend) were included in Model 6. Methods to control for multiple testing were not used here. We were interested in the change in the effect size of the association between functional health literacy and mortality following the inclusion of various cognitive, sociodemographic and health variables. In the Results section of the main text here, only the HRs and 95% CIs for the functional health literacy measures are reported. A more detailed

**Table 1** Participant characteristics for participants alive or dead at censoring date and p values to determine whether these characteristics differed by survival status

| | n | Alive | Dead | P values |
|---|---|---|---|---|
| Survival time (years), mean (SD) | 795 | 8.19 (0.66) | 5.23 (2.14) | |
| Age (years) at wave 2, mean (SD) | 795 | 72.54 (0.70) | 72.41 (0.72) | 0.068 |
| Sex, n (%) | 795 | | | 0.069 |
| Male | | 336 (50.5) | 77 (59.2) | |
| Female | | 329 (49.5) | 53 (40.8) | |
| Years of education, mean (SD) | 795 | 10.80 (1.16) | 10.71 (1.10) | 0.417 |
| Occupational class, n (%) | 780 | | | 0.001 |
| Professional | | 142 (21.7) | 12 (9.4) | |
| Managerial/technical | | 249 (38.1) | 49 (38.6) | |
| Skilled: non-manual | | 140 (21.4) | 26 (20.5) | |
| Skilled: manual | | 96 (14.7) | 35 (27.6) | |
| Manual | | 26 (4.0) | 5 (3.9) | |
| Self-reported health, n (%) | 795 | | | <0.001 |
| Poor/fair | | 47 (7.1) | 26 (19.9) | |
| Good | | 195 (29.4) | 40 (30.5) | |
| Very good/excellent | | 422 (63.5) | 65 (49.6) | |
| HADS total, mean (SD) | 794 | 7.02 (4.37) | 7.42 (4.62) | 0.342 |
| Townsend Disability Scale, mean (SD) | 794 | 0.89 (1.82) | 1.60 (2.48) | 0.001 |

HADS, Hospital Anxiety and Depression Scale.

## RESULTS

A total of 796 participants completed the functional health literacy measures at wave 2 (figure 1). Following removal of one participant without information on date of death, 130 participants had died, and 665 participants were alive at the censoring date. Participant characteristics are reported in table 1, and functional health literacy

**Table 2** Mean scores (SD) on measures of functional health literacy and cognitive ability by survival status and p values to determine whether these scores differ by survival status

| | n | Alive | Dead | P values |
|---|---|---|---|---|
| REALM score | 794 | 65.08 (2.39) | 64.67 (3.02) | 0.015 |
| S-TOFHLA score | 744 | 38.00 (3.85) | 36.69 (5.37) | 0.025 |
| NVS score | 789 | 2.92 (1.90) | 2.48 (1.92) | 0.011 |
| General functional health literacy | 740 | 0.05 (0.98) | −0.24 (1.08) | 0.007 |
| Age-11 IQ | 752 | 101.08 (14.99) | 98.55 (16.33) | 0.091 |
| Current fluid ability | 789 | 0.07 (0.99) | −0.34 (1.00) | <0.001 |

NVS, Newest Vital Sign; REALM, Rapid Estimate of Adult Literacy in Medicine; S-TOFHLA, Shortened Test of Functional Health Literacy in Adults.

description of the results for all variables in the models is given in the online supplementary materials.

and cognitive ability scores are shown in table 2. Those who died were more likely to be from a lower occupational class, were more likely to report poorer health and reported more disability than those who survived. Participants who had died had lower scores on all the functional health literacy measures and had lower fluid cognitive ability scores in older age. Age-11 IQ did not differ between the two groups.

Table 3 shows the rank-order correlations between functional health literacy and cognitive ability measures. These have been reported elsewhere.[23][25] The three functional health literacy measures correlated moderately with each other ($r$=0.35–0.44, p<0.001), and higher scores on the functional health literacy measures were correlated with higher age-11 IQ ($r$=0.44–0.51, p<0.001) and higher fluid ability ($r$=0.38–0.55, p<0.001). The three functional health literacy measures tended to correlate more strongly with measures of cognitive ability than with each other. The general functional health literacy measure also showed a strong positive correlation with both age-11 IQ ($r$=0.61, p<0.001) and fluid ability in older age ($r$=0.63, p<0.001). The correlations between all variables examined in this analysis are reported in online supplementary table 1.

The HRs for the association between functional health literacy and mortality are shown in table 4. HRs for all variables entered into the models are reported in online supplementary tables 2–5. In all models, the assumptions of proportional hazards were met. Given the high correlations between functional health literacy and cognitive

**Table 3**  Rank-order correlations between functional health literacy and cognitive ability measures

|  | 1 | 2 | 3 | 4 | 5 | 6 |
|---|---|---|---|---|---|---|
| 1 REALM | – |  |  |  |  |  |
| 2 S-TOFHLA | 0.40* | – |  |  |  |  |
| 3 NVS | 0.35* | 0.44* | – |  |  |  |
| 4 General functional health literacy | 0.71* | 0.80* | 0.78* | – |  |  |
| 5 Age-11 IQ | 0.44* | 0.48* | 0.51* | 0.61* | – |  |
| 6 Current fluid ability | 0.38* | 0.55* | 0.55* | 0.63* | 0.57* | – |

*P<0.001.
NVS, Newest Vital Sign; REALM, Rapid Estimate of Adult Literacy in Medicine; S-TOFHLA, Shortened Test of Functional Health Literacy in Adults.

ability, variance inflation factors (VIFs) were calculated to check for multicollinearity. VIF values for all models were low (highest VIF=2.15), suggesting that there was no multicollinearity in these models.

### REALM
The HRs for the REALM represent the risk of dying for a one-point increase in the REALM (max score=66). The REALM did not significantly predict mortality in Model 1 (HR 0.95, 95% CI 0.90 to 1.01) adjusting for age and sex or subsequently with the addition of education (Model 2), age-11 IQ (Model 3), fluid ability (Model 4), occupational class (Model 5) or health status (Model 6).

### S-TOFHLA
The HRs for the S-TOFHLA represent the risk of mortality for a one-point increase in S-TOFHLA score (max score=40). With age and sex controlled for, a one-point increase in S-TOFHLA reduced the risk of dying by 5% (Model 1 HR 0.95, 95% CI 0.92 to 0.98). Inclusion of education (Model 2) and age-11 IQ (Model 3) did not attenuate this association. This association was attenuated and became non-significant in Model 4 with the inclusion of fluid ability (HR 0.97, 95% CI 0.93 to 1.01) and remained non-significant and continued to reduce in size

following the addition of occupational class (Model 5) and health status (Model 6).

### NVS
The HRs for NVS represent the risk of mortality for a one-point increase in NVS score (max score=6). In Model 1, in which age and sex were entered as covariates, NVS significantly predicted mortality. A one-point increase in NVS score reduced the risk of dying by 12% (HR 0.88, 95% CI 0.80 to 0.97). The addition of years of education (Model 2) did not attenuate this association. Age-11 IQ was added in Model 3, and this did little to change the association between NVS and mortality. The inclusion of fluid ability in Model 4 greatly attenuated the association between NVS and mortality, and this association became non-significant (HR 0.96, 95% CI 0.86 to 1.08). This association remained non-significant following the inclusion of occupational class (Model 5) and health status variables (Model 6).

### General functional health literacy
The HRs for general functional health literacy represent the risk of mortality for a one SD increase in general functional health literacy. General functional health literacy predicted mortality in Model 1, controlling for age and

**Table 4**  HRs (95% CIs) for the association between four measures of functional health literacy and mortality, controlling for sociodemographic, cognitive and health variables

|  | Model 1 Age and sex | Model 2 +education | Model 3 +age-11 IQ | Model 4 +current fluid ability in older age | Model 5 +occup class | Model 6 +health status |
|---|---|---|---|---|---|---|
| REALM | 0.95 (0.90 to 1.01) n=794 | 0.96 (0.90 to 1.01) n=794 | 0.96 (0.90 to 1.02) n=752 | 0.97 (0.91 to 1.04) n=746 | 0.97 (0.90 to 1.04) n=731 | 1.00 (0.92 to 1.07) n=728 |
| S-TOFHLA | 0.95 (0.92 to 0.98)** n=744 | 0.95 (0.92 to 0.98)** n=744 | 0.95 (0.91 to 0.98)** n=702 | 0.97 (0.93 to 1.01) n=697 | 0.98 (0.94 to 1.02) n=682 | 1.00 (0.95 to 1.05) n=680 |
| NVS | 0.88 (0.80 to 0.97)** n=789 | 0.88 (0.80 to 0.97)* n=789 | 0.89 (0.80 to 0.99)* n=746 | 0.96 (0.86 to 1.08) n=742 | 0.97 (0.86 to 1.09) n=727 | 0.96 (0.85 to 1.08) n=724 |
| General functional health literacy | 0.77 (0.65 to 0.92)** n=740 | 0.75 (0.61 to 0.90)** n=740 | 0.74 (0.59 to 0.93)* n=698 | 0.87 (0.67 to 1.13) n=694 | 0.911 (0.70 to 1.19) n=679 | 0.95 (0.72 to 1.25) n=677 |

*P<0.05, **p<0.01.
NVS, Newest Vital Sign; occup class, occupational class; REALM, Rapid Estimate of Adult Literacy in Medicine; S-TOFHLA, Shortened Test of Functional Health Literacy in Adults.

sex. A one SD increase in general functional health literacy reduced the risk of mortality by 23% (HR 0.77, 95% CI 0.65 to 0.92). Including years of education in Model 2 and age-11 IQ in Model 3 did little to change the association between general functional health literacy and mortality. Current fluid ability was included in Model 4, and this attenuated the association between general functional health literacy and mortality, and this association was no longer significant (HR 0.87, 95% CI 0.67 to 1.13). Adding occupational social class in Model 5 did little to change the association between general functional health literacy and mortality. Health status variables were added in Model 6, and the association between general functional health literacy and mortality was further attenuated and remained non-significant.

All models were rerun using only participants who had complete data on all of the variables of interest. These models are shown in online supplementary tables 6–9. The associations between functional health literacy and mortality were similar to those reported here, except that, in Model 1 for the REALM (online supplementary table 6), higher scores on the REALM significantly reduced the risk of mortality. This association was no longer significant in Model 2, following the inclusion of age-11 IQ.

## Sensitivity analyses

Participants who may have a dementia or possible pathological cognitive impairment were not removed prior to running these analyses. One participant self-reported a diagnosis of dementia at the wave 2 assessment. Five participants in this sample have Mini-Mental State Examination scores below the often used cut-off of 24[45] (one participant scored 20/30, one scored 22/30 and three scored 23/30), which suggests a possible cognitive impairment. To determine whether the presence of dementia or possible cognitive impairment affects the results, these analyses were rerun excluding these six individuals. All associations were very similar to those reported above (results not shown; available from the authors), and therefore, the presence of dementia or possible cognitive impairment did not affect the main results.

## DISCUSSION

This study investigated whether prior cognitive ability measured in childhood and current fluid cognitive ability measured in older adulthood played different roles in the association between functional health literacy and mortality. Three measures of functional health literacy were used; the REALM, S-TOFHLA and NVS. These three measures were also used to create a composite measure of functional health literacy. The REALM, a test that requires only the ability to read and correctly pronounce medical words, did not predict mortality, even in minimally adjusted models (though it had a slightly stronger and significant association when only those with full data were included, as shown in online supplementary analysis). When using functional health literacy tests that assessed reading comprehension and numeracy (S-TOFHLA, NVS and general functional health literacy), functional health literacy predicted mortality in models adjusting for age, sex and education only. Individuals who had higher scores on the S-TOFHLA, NVS and general functional health literacy had a lower risk of mortality than those with lower scores. Accounting for prior intelligence measured in childhood did not change this association. The association between functional health literacy and mortality disappeared when contemporaneous fluid ability was accounted for. The attenuation was particularly large for the NVS and general functional health literacy.

Two previous studies used functional health literacy tests that measure reading comprehension and numeracy to investigate the role that cognitive function plays in the association between functional health literacy and mortality.[32 33] These studies measured cognitive function concurrently with health literacy in middle-age or older adulthood and found that, although the size of the association between functional health literacy and mortality was reduced, functional health literacy still predicted mortality when cognitive function was controlled for.[32 33] We investigated the role that both childhood cognitive ability and cognitive ability in older age have on the association between functional health literacy and mortality. Here, fluid ability, but not childhood intelligence, attenuated the association between functional health literacy and mortality such that the association was no longer significant. Childhood cognitive ability, which was measured decades prior to the functional health literacy assessment, is thought to reflect the relatively stable trait of lifelong intelligence, whereas current fluid ability, which was measured when participants were approximately 73 years old, is a measure of current cognitive competence.[23] These results suggest that, whereas childhood intelligence did not play a role in the association between functional health literacy and mortality, current fluid-type cognitive ability in older adulthood accounted for a large proportion of this association.

A strength of this current study is that detailed measures of cognitive ability were used. Childhood intelligence was measured using a standardised test of intelligence which had good concurrent validity with other intelligence tests.[35] The fluid ability measure comprised many standardised neuropsychological tests. Both Baker et al[32] and Bostock and Steptoe[33] used brief measures of cognitive function. Baker et al[32] used specific items from the Mini-Mental State Examination, a measure designed to screen for cognitive impairment[45] which is insensitive to individual differences in healthy cognitive ageing. Bostock and Steptoe[33] used three brief cognitive tests administered in a non-standardised way in the participants' own home. These studies may not have used tests sensitive enough, or that covered a necessary range of cognitive functions, to fully account for the association between health literacy and mortality.

Another advantage of the current study is the use of three different tests of functional health literacy. All tests

were used to measure functional health literacy; however, each test required the participant to carry out different health-related tasks. Whereas the REALM required the participant only to read and correctly pronounce words, the S-TOFHLA and NVS are more cognitively demanding tasks that assessed both reading comprehension and numeracy skills. Using these three measures enabled us to investigate whether different patterns of association between functional health literacy and mortality were found when using the different tests. By using three measures of functional health literacy, we were also able to create a composite measure of functional health literacy. This general measure was derived with the aim of creating a score that captures the shared variance between the three functional health literacy tests, providing a more comprehensive measure of functional health literacy.

The results of this study support the proposal by Reeve and Basalik[26] that functional health literacy may not be a unique construct; instead, it is tenable that tests of functional health literacy may in fact be largely measuring cognitive ability. First, we found, as has been reported elsewhere,[23 25] that tests of health literacy tended to correlate more strongly with tests of cognitive ability than with each other. The original paper describing the S-TOFHLA found that this test correlated with the REALM at $r=0.80$,[6] suggesting that these tests are measuring the same underlying ability. However, other studies have found moderate correlations between these tests, similar to ours.[46] Second, we found that the NVS, S-TOFHLA and general functional health literacy no longer predicted mortality when accounting for fluid cognitive ability. The results of our study suggest that health literacy may not be independent of cognitive ability. This attenuation is likely to be because there is an overlap in the content of tests of fluid ability and the NVS and S-TOFHLA. The NVS and S-TOFHLA are cognitively demanding tasks that are likely to be substantially measuring fluid-type cognitive abilities, such as working memory and reasoning, that decline with increasing age.[15] Childhood cognitive ability did not attenuate the association between functional health literacy and mortality, suggesting that the NVS and S-TOFHLA are measuring current fluid-type cognitive capability in old age, and not lifelong intelligence. Current fluid ability in older age may be driving much of the association between functional health literacy and mortality largely because tests of functional health literacy are assessing mostly the same underlying abilities as measures of fluid ability.

Some researchers have questioned the validity of some of the functional health literacy tests used here. The Test of Functional Health Literacy in Adults is often reported as the gold standard functional health literacy test.[39] However, the NVS has been found to have poor concurrent validity with the Test of Functional Health Literacy in Adults.[39] In support of this, we found that the rank-order correlation between the NVS and S-TOFHLA was modest ($r=0.44$). Concerns have been raised about the fact that the REALM assesses only the ability to read and

pronounce words.[38] Knowing how to pronounce medical words may not be directly related to the ability to understand medical information, and therefore, this may not adequately cover all the domains of functional health literacy.[38] Indeed, all the tests used here were designed to largely measure the component of health literacy known as functional health literacy. None of these measures assess other components of health literacy such as the skills required to critically analyse health information or the communicative skills needed to participate and navigate in the healthcare environment.[3] Assessments of health literacy that are designed to measure a much broader range of health literacy skills are available, such as the Health Literacy Questionnaire (HLQ)[47] and the European Health Literacy Survey Questionnaire (HLS-EU-Q).[48] The HLQ assesses nine dimensions of health literacy, including the ability to actively manage health and navigate the healthcare system.[47] Whereas the HLS-EU-Q measures self-reported skills in being able to access, understand, appraise and apply health-related information in the healthcare setting, as well as in disease prevention and health promotion.[48] Fluid cognitive ability may not play a role in the association between health literacy and mortality if we used these self-reported, broad, measures of health literacy, rather than the objective, but narrow, functional health literacy tests used here.

There are some limitations to this study. The LBC1936 participants were followed up for the first time at age 70 years, and therefore, the sample used in this analysis will likely suffer from a survival bias as this sample is made up of individuals who have survived to the age of 70 years. LBC1936 participants also tended to have higher scores on the MHT (age-11 IQ test) than Scottish-wide and Edinburgh-wide participants who also sat this test in 1947 as part of the Scottish Mental Survey.[36] Thus, individuals in this sample tended to be brighter than the original Scottish Mental Survey 1947 participants. This analysis only examined the association between functional health literacy and all-cause mortality. It is possible that there are different relationships between functional health literacy and cause-specific mortality, for example, functional health literacy may only predict deaths linked to unhealthy lifestyles, such as cardiovascular disease. The follow-up period in this study was relatively short, and therefore, only a small percentage of participants had died. Future studies should investigate mortality over a longer follow-up period and in larger samples to examine whether there are different patterns of association between functional health literacy and cause-specific mortality.

We investigated whether childhood cognitive ability and fluid ability in older age play independent roles in the association between functional health literacy and mortality. The results indicate that fluid-type cognitive capability may account for the association between functional health literacy and mortality, whereas childhood cognitive ability—an indicator of lifelong intelligence—does not. Researchers and clinicians should be

aware that lower functional health literacy scores may actually reflect lower cognitive ability in older age, and that current cognitive capacity in older adulthood, but not lifelong intelligence, may be driving the association between functional health literacy and mortality. Future research examining the association between functional health literacy and mortality, and other health indicators, should also include measures of cognitive ability to be able to properly disentangle the relationship between functional health literacy and health.

**Acknowledgements** We thank the LBC1936 participants and LBC1936 research team members.

**Contributors** CF-R discussed and planned the study and analyses, analysed the data, interpreted the data and drafted the initial manuscript. JMS discussed and planned the study and analyses, interpreted the data and contributed to the manuscript. IJD discussed and planned the study and analyses, interpreted the data and contributed to the manuscript.

**Funding** This work was supported by the University of Edinburgh Centre for Cognitive Ageing and Cognitive Epidemiology, part of the cross council Lifelong Health and Wellbeing Initiative, funded by the Biotechnology and Biological Sciences Research Council (BBSRC) and Medical Research Council (MRC) (grant no MR/K026992/1). The Lothian Birth Cohort 1936 (LBC1936) is supported by Age UK (Disconnected Mind project).

**Competing interests** None declared.

**Patient consent** Not required.

**Ethics approval** Ethical approval was obtained from the Scotland A Research Ethics Committee (07/MRE00/58).

**Provenance and peer review** Not commissioned; externally peer reviewed.

**Data sharing statement** Lothian Birth Cohort 1936 data can be requested from the Lothian Birth Cohort 1936 research team, following completion of a data request application. More information can be found online (http://www.lothianbirthcohort.ed.ac.uk/content/collaboration).

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
