## [Reviewer comments · BMJ Open]

ARTICLE DETAILS

TITLE (PROVISIONAL)	The role of cognitive ability in the association between functional health literacy and mortality in the Lothian Birth Cohort 1936: a prospective cohort study
AUTHORS	Fawns-Ritchie, Chloe; Starr, John; Deary, Ian

VERSION 1 – REVIEW

REVIEWER	Veronique Roger Mayo Clinic, USA
REVIEW RETURNED	23-Mar-2018

GENERAL COMMENTS	The paper by Fawns-Ritchie et al. examines the association between intelligence and health literacy with mortality. The design is a prospective study, and the authors use data from the Lothian Birth Cohort 1936 study which recruited 1091 older adults born in 1936 in Scotland. The children were all born in 1936 and they underwent intelligence testing in 1947. The present study focuses on the sample of individuals who were followed up for the first time at age 70. The present study uses a subsample of 795 individuals who completed tests of health literacy when they were approximately 73 years of age. The results demonstrate that higher health literacy was associated with the lower risks of death adjusted for age and sex. Adjusting for fluid-type cognitive ability attenuated the associations between health literacy and mortality. The study of health literacy is an important and timely topic as it is increasingly being recognized that navigating the high complexity healthcare environment that characterizes contemporary delivery of care, requires making complex decisions, being able to self-manage, and choose health related behaviors. All of these activities do require to understand health information, and hence the purported importance of health literacy in outcomes. The Lothian Birth Cohort of 1936 has already been studied and used to demonstrate the associations between health literacy and measure the physical health vs. physical fitness, number of natural teeth and body mass index. Measures of cognitive ability attenuated these relationships. The present paper focuses on mortalities and outcome to again examine the associations between health literacy and mortality while examining the possible mediations by measures of childhood intelligence and health cognition. With the importance of the topic, it is important to underscore that the paper addresses a similar question as the one previously mentioned, albeit with a different outcome. Therefore the innovative
--

	nature of the study is modest. The cohort (Birth Cohort in 1936) was studied again for the first time at age 70 which creates a survival bias that should be acknowledged. The processes and approach leading to the generation of the data should be clarified by incorporation of a flow diagram. The authors emphasize the importance of having four measures of health literacy but this particular feature is underexploited in the results. The authors do not explicitly discuss the possible diverseness of dementia which is likely to, if present, confound substantially the findings reported. The overarching finding of this study is that associations between health literacy and mortality are more attenuated by adjusting for cognitive ability. While the reviewers acknowledge the importance of the topic, the design of the study comes across as confusing, and the findings of marginal interest as they do underrepresent a novel finding. Adding a conceptual framework or diagram in the paper would improve its credibility to relate to the findings.
--	--

REVIEWER	Richard Osborne Deakin University, Australia
REVIEW RETURNED	28-Apr-2018

GENERAL COMMENTS	bmj open 2018 022502 thank you for the opportunity to review this manuscript. The research explores the association between an aspect of health literacy (HL) and survival in a robust cohort study. This is one of the only cohorts that have HL as a measure hence it is important data to explore. I have a number of methodological and conceptual concerns with the paper that need to be attended to. There is a logical inconsistency regarding the theory, definition and operationalization of health literacy. A multidimensional definition is proposed, which includes p4 l6 “capacity to obtain, process and understand basic health information and services needed to make basic health decisions.”, yet the tools used to measure HL are unidimensional – focused on health-related reading and numeracy ability, i.e., elements directly related to obtaining and processing information are omitted. A reader would be misled by how the HL is portrayed in this paper. Health-related reading and numeracy ability is widely described as functional health literacy. There are other components of HL that have not been measured. P5l9 I agree that it makes sense to consider that the functional health literacy part of the broader multi-dimensional HL construct might be related to cognitive function. I encourage the authors to explain their narrative in the Introduction in terms of effect sizes and not imply the relative importance of associations in terms of p-values (i.e., significance tests). Methods The three measures of HL used in this study are all unidimensional –
---

	focused on various aspects of functional health literacy. The proposed 'general health literacy measure' is therefore a composite of functional health literacy. There is a considerable literature that questions the robustness of the REALM even in the country from which it was developed (USA). Purpose built multi-dimensional health literacy tools do exist, are applied widely in UK/EU, and cover 3 to 9 dimensions, e.g., the HLS-EU and the HLQ, respectively. The TOFHLA is a timed reading comprehension tests that use the modified Cloze procedure, in which every 5th to 7th word in a passage is omitted and replaced with a blank space. The respondent must select a word to fit into the blank spaces from the 4 multiple-choice options provided for each space. Given that it is a timed test, and requires substantial cognitive capacity, I think it is important that the reader is provided with this information in this setting. The nature of the test, the cognitive challenge, is likely to be a reason for its association with fluid ability. The same goes for the NVS, but to a smaller extent. Given that education is a well-known correlate of health literacy (a potential confounder) I do not understand why this isn't included in model 2 onward. This needs to be part of the basic model. Results Reporting correlations to 3 decimal places seems like spurious accuracy. The inclusion of the so called General health literacy, i.e., the composite of the 3 measures, and little to the granularity or robustness of the data analysis. Remove the redundant 1.00 in table 3. Discussion A simple content analysis and structural analysis of the HL and cognitive tests used seems prudent. Rather than the content of the HL measures, it may be the kinds of tasks required to complete the tests leads to the strong association between HL and fluid ability. The discussion should outline the limitations of the HL tools used, especially the coverage of one of the three broad areas of HL, i.e., functional health literacy, rather than the many other domains that represent the full construct.
--	--

VERSION 1 – AUTHOR RESPONSE

Reviewer 1

1. With the importance of the topic, it is important to underscore that the paper addresses a similar question as the one previously mentioned, albeit with a different outcome. Therefore the innovative nature of the study is modest.

Response: We thank reviewer 1 for their helpful comments. We agree with reviewer 1 that this is an important topic to investigate. Whereas we agree that this paper is complementary to that by Möttus et al. (2012, *Health Psychol*, 33:164-73), which investigated the role that cognitive ability

plays in the association between health literacy and physical health using the same sample as used in the current study (the Lothian Birth Cohort 1936), we judge that our paper contributes a different and important topic. Möttus et al. (2012) examined the cross-sectional association between health literacy, older-age cognitive ability, and physical health. Our study investigated a different health outcome; death. We think that mortality is an especially important health outcome to investigate and is important to address in addition to physical health and fitness. While Möttus et al. (2012) looked at the cross-sectional association between health literacy, old-age cognitive ability, and physical health, our study is a prospective study in which Lothian Birth Cohort 1936 participants were followed up for 8 years to determine mortality status.

We have edited the introduction to make the reader fully aware of the similarities between this paper and Möttus et al.'s (2012) paper.

Page 6: *"We note that this is the same sample as used in Möttus et al.[25] to investigate the association between functional health literacy, cognitive ability and physical health. In this previous study,[25] physical health was measured concurrently with fluid ability and functional health literacy."*

We have edited the introduction to make the reader aware that we believe this current paper is unique from Möttus et al. (2012).

Page 6: *"The current analysis is different from and complementary to this previous study in that we followed up the participants for 8 years to determine mortality status—obviously a most important health outcome."*

This makes it clear to the reader that our outcome is different and is also the product of many years of longitudinal follow-up, unlike the study by Möttus et al. (2012).

We have also edited the introduction to make it clear to the reader that our analysis is also unique from Baker et al. (2008, *J Gen Intern Med*, 23:723-6) and Bostock and Steptoe (2012, *BMJ*, 344:e1602) who investigated the role of cognitive function in the relationship between health literacy and mortality.

Page 6: *"Studies that have examined the role that cognitive function plays in the association between functional health literacy and mortality used brief cognitive measures collected at the same time as the functional health literacy tests.[32, 33] It is not known whether early life cognitive ability and cognitive ability in older age play different roles in the association between health literacy and mortality. The current analysis utilises cognitive test scores collected in childhood, which are thought to measure the trait of lifelong intelligence, and current cognitive ability in older age, measured at approximately 73 years and contemporaneously with functional health literacy."*

2. The cohort (Birth Cohort in 1936) was studied again for the first time at age 70 which creates a survival bias that should be acknowledged.

Response: We agree with reviewer 1 that there may be a survival bias because the Lothian Birth Cohort participants were tested again for the first time at age 70 years. We have therefore updated the discussion to highlight to the reader that a survival bias may be present in our analysis. We have also updated the discussion to highlight that the LBC1936 participants tended to be brighter than the Scottish-wide participants in the Scottish Mental Survey 1947.

Page 23: *"There are some limitations to this study. The LBC1936 participants were followed-up for the first time at age 70 years and therefore the sample used in this analysis will likely suffer from a survival bias as this sample is made up of individuals who have survived to the age of 70 years. LBC1936 participants also tended to have higher scores on the Moray House Test (age-11 IQ test) than Scottish-wide and Edinburgh-wide participants who also sat this test in 1947 as part of the Scottish Mental Survey.[36] Thus, individuals in this sample tended to be brighter than the original Scottish Mental Survey 1947 participants."*

3. The processes and approach leading to the generation of the data should be clarified by incorporation of a flow diagram.

Response: We thank reviewer 1 for this helpful suggestion. A flow diagram detailing how the analytic sample for this study was derived has been added to the manuscript (see Figure 1).

4. The authors emphasize the importance of having four measures of health literacy but this particular feature is underexploited in the results.

Response: We think that the use of the four health literacy measures is a real strength of this paper. Within the results section, we report each of the health literacy measures in turn. We acknowledge that we had not specifically discussed the different patterns of association between health literacy and mortality when using the different tests of health literacy. Therefore, we have updated the discussion to highlight to readers the use of the four measures of health literacy, how these measures differ from each other, and the different pattern of results that were found using these different tests.

We have updated the first paragraph in the discussion to emphasise that this study used four different measures of health literacy.

Page 19: *“Four measures of functional health literacy were used; the REALM, S-TOFHLA, NVS, and a composite measure of functional health literacy.”*

We highlight to the reader that different patterns of association with mortality were found when using the REALM, which requires only the ability to read and pronounce words, than when we used the S-TOFHLA, NVS and the general functional health literacy measure, in which reading comprehension and numeracy are assessed.

Page 19: *“The REALM, a test that requires only the ability to read and correctly pronounce medical words, did not predict mortality, even in minimally adjusted models (though it had a slightly stronger and significant association when only those with full data were included, as shown in supplementary analysis). When using functional health literacy tests that assessed reading comprehension and numeracy (S-TOFHLA, NVS, and general functional health literacy), functional health literacy predicted mortality in models adjusting for age, sex and education only.”*

We have also added a section to the discussion detailing the advantages of using multiple measures of health literacy. First, this meant we could compare the results of tests that assess only reading and pronunciation skills (REALM) and tests that assess more cognitively demanding reading comprehension and numeracy (S-TOFHLA and NVS).

Page 21: *“Another advantage of the current study is the use of three different tests of functional health literacy. All tests were used to measure functional health literacy; however, each test required the participant to carry out different health-related tasks. Whereas the REALM required the participant only to read and correctly pronounce words, the S-TOFHLA and NVS are more cognitively demanding tasks that assessed both reading comprehension and numeracy skills. Using these three measures enabled us to investigate whether different patterns of association between functional health literacy and mortality were found when using the different tests.”*

Second, we highlight that using multiple measures enabled us to create a general measure of functional health literacy which captures the shared variance between the measures.

Page 21: *“By using three measures of functional health literacy, we were also able to create a composite measure of functional health literacy. This general measure was derived with the aim of creating a score that captures the shared variance between the three functional health literacy tests, providing a more comprehensive measure of functional health literacy.”*

5. The authors do not explicitly discuss the possible diverseness of dementia which is likely to, if present, confound substantially the findings reported.

Response: The Lothian Birth Cohort 1936 is a sample designed to assess healthy cognitive ageing (Deary et al., 2007, 2012). The cognitive assessment that is given as part of the Lothian Birth Cohort 1936 is lengthy and cognitively demanding. The tests administered are designed for use in healthy older adults. Therefore, the participants in the Lothian Birth Cohort 1936 are generally cognitively healthy. Because of this, in the main analyses, we did not explicitly remove participants who self-reported a diagnosis of dementia, or who had MMSE scores that indicated a cognitive impairment.

We have updated the methods section to emphasise that the Lothian Birth Cohort 1936 is a study of healthy, non-pathological, older individuals.

Page 7: *“LBC1936 was designed principally to investigate healthy, non-pathological, cognitive ageing.”*

Following reviewer 1’s comments, we investigated whether any participants used in the current analyses self-reported a diagnosis of dementia at wave 2 or had an MMSE score below the often-used cut-off of 24 at wave 2. One participant self-reported a diagnosis of dementia at wave 2, and 5 participants had an MMSE score below 24 (1 participant score 20/30, 1 scored 22/30, and 3 scored 23/30). To determine whether the presence of a possible cognitive impairment or dementia in this very small number of the participants affected the findings, we re-ran the analyses after removing these 6 individuals. All associations were very similar and therefore the presence of a possible pathological cognitive impairment or dementia did not affect the results of this study. We have updated the results to include this sensitivity analyses.

Page 19: *“Sensitivity analyses: Participants who may have a dementia or possible pathological cognitive impairment were not removed prior to running these analyses. One participant self-reported a diagnosis of dementia at the wave 2 assessment. Five participants in this sample have mini-mental state exam scores below the often-used cutoff of 24[45] (one participants scored 20/30, one scored 22/30 and three scored 23/30), which suggests a possible cognitive impairment. To determine whether the presence of dementia or possible cognitive impairment affects the results, these analyses were re-run excluding these 6 individuals. All associations were very similar to those reported above (results not shown; available from the authors) and therefore the presence of dementia or possible cognitive impairment did not affect the main results.”*

6. While the reviewers acknowledge the importance of the topic, the design of the study comes across as confusing, and the findings of marginal interest as they do underrepresent a novel finding. Adding a conceptual framework or diagram in the paper would improve its credibility to relate to the findings.

Response: As outlined above, we judge that this study does represent a novel finding, and we have updated the manuscript accordingly. Significant changes have been made to the introduction and discussion to make the design and importance of the study clearer.

Changes to introduction: We have updated the introduction to more clearly show what previous research has found and what our study adds to this.

We provide detailed information about what Möttus et al. (2012) found when investigating the role that childhood and old age cognitive ability play in the association between health literacy and physical health.

- **Page 5:** *“A study using participants from the Lothian Birth Cohort 1936[25]—the same sample used in the current study—investigated whether cognitive ability in childhood and late adulthood attenuated the association between functional health literacy and physical health. In models without cognitive function, functional health literacy was associated with all three of the measures of physical health assessed. Addition of cognitive ability in older age significantly attenuated the association between functional health literacy with physical fitness by 43%, and number of natural teeth by 39%; however, it did not attenuate the association between functional health literacy and body mass index (BMI). Conversely, whereas childhood cognitive ability did not attenuate the association between functional health literacy and physical fitness, it attenuated the association between functional health literacy and number of teeth by 30%, and BMI by 88%. In the fully adjusted model which included childhood and late adulthood cognitive ability, as well as other early-life factors, the association between functional health literacy and physical fitness, though attenuated by 43%, remained significant,[25] suggesting that functional health literacy may play a small but unique role in physical fitness.”*
- After reporting what Baker et al. (2008) and Bostock and Steptoe (2012) found when investigating the role of cognitive function in the association between health literacy and mortality, we highlight to the reader that these previous studies have only used cognitive

function tests measured at the same time as the health literacy assessments in either middle-age or old-age. No one has investigated the role that both early life and older age cognitive function play in this association. This is important, because cognitive function from early life assesses the life-long trait, whereas cognitive function scores in older age capture both that trait and any age-related decline that has taken place.

Page 6: *“Studies that have examined the role that cognitive function plays in the association between functional health literacy and mortality used brief cognitive measures collected at the same time as the functional health literacy tests.[32, 33]”*

- We then state that the role of both childhood and old age cognitive ability in the association between health literacy and mortality has not been investigated.

Page 6: *“It is not known whether early life cognitive ability and cognitive ability in older age play different roles in the association between health literacy and mortality.”*

- We then explicitly state the aim of the current study.
- **Pages 6-7:** *“The aim of this study was to determine whether childhood cognitive ability and current cognitive ability in older adulthood play unique roles in the association between functional health literacy and mortality.”*

We think this now makes the reason for carrying out this study explicitly clear to readers.

Changes to discussion: We have made considerable changes to the discussion so that it is clear to the reader exactly what we did and how our results fit into the wider literature on a) the association between health literacy and cognitive ability, and b) the role of cognitive ability in the association between health literacy and mortality. We did this in the following ways:

- We start this discussion by explicitly stating what was done.
Page 19: *“This study investigated whether prior cognitive ability measured in childhood and current fluid cognitive ability measured in older adulthood played different roles in the association between functional health literacy and mortality.”*
- We compare our results to previous studies investigating the role of cognitive function in the association between health literacy and mortality. We again highlight to the reader that our study is unique in that we were able to investigate the role that both childhood and old age cognitive ability play in the association between health literacy and mortality.
Page 20: *“Two previous studies used functional health literacy tests that measure reading comprehension and numeracy to investigate the role that cognitive function plays in the association between functional health literacy and mortality.[32, 33] These studies measured cognitive function concurrently with health literacy in middle-age or older adulthood and found that, although the size of the association between functional health literacy and mortality was reduced, functional health literacy still predicted mortality when cognitive function was controlled for.[32, 33] We investigated the role that both childhood cognitive ability, and cognitive ability in older age have on the association between functional health literacy and mortality.”*
- We state that our findings are also in support of Reeve and Basalik’s proposal that health literacy and cognitive function are measuring the same underlying construct.
Page 21: *“The results of this study support the proposal by Reeve and Basalik[26] that functional health literacy may not be a unique construct; instead, it is tenable that tests of functional health literacy may in fact be measuring cognitive ability. Here, NVS, S-TOFHLA and general functional health literacy no longer predicted mortality when accounting for fluid ability.”*
- We then use Reeve and Basalik’s proposal to explain our findings. We propose that the reason the association between health literacy and mortality is attenuated when accounting for fluid-type cognitive ability is because there is an overlap in the content of the tests, and the health literacy tests that assess reading comprehension and numeracy are to a substantial extent measuring fluid-type abilities such as working memory and reasoning.
Pages 21-22: *“This attenuation is likely to be because there is an overlap in the content of tests of fluid ability and the NVS and S-TOFHLA. The NVS and S-TOFHLA are cognitively demanding tasks that are likely to be substantially measuring fluid-type cognitive abilities, such as working memory and reasoning, that decline with increasing age.[15]”*

These substantial changes made to the introduction and discussion now make the design of the study and the rationale for carrying out the study much clearer.

Reviewer 2

1. There is a logical inconsistency regarding the theory, definition and operationalization of health literacy. A multidimensional definition is proposed, which includes p4 l6 “capacity to obtain, process and understand basic health information and services needed to make basic health decisions.”, yet the tools used to measure HL are unidimensional – focused on health-related reading and numeracy ability, i.e., elements directly related to obtaining and processing information are omitted. A reader would be misled by how the HL is portrayed in this paper. Health-related reading and numeracy ability is widely described as functional health literacy. There are other components of HL that have not been measured.

Response: We thank reviewer 2 for their comments. On reflection, we agree with the reviewer that we need to make it clear to the reader that the measures used are assessing functional health literacy. We have updated the manuscript to reflect this. We now use the term “functional health literacy” throughout, including in the title.

We have also edited the introduction to reflect that functional health literacy is only one component of health literacy.

Page 4: *“One component of health literacy is functional health literacy—the reading, writing, and numeracy skills required to understand health information. [3, 5, 6]”*

We have updated the discussion to highlight to readers that the tests used in the current study mostly assess functional health literacy and do not measure many other components of health literacy.

Page 22: *“Indeed, all the tests used here were designed to largely measure the component of health literacy known as functional health literacy. None of these measures assess other components of health literacy such as the skills required to critically analyse health information or the communicative skills needed to participate and navigate in the health care environment.[3]”*

2. I encourage the authors to explain their narrative in the Introduction in terms of effect sizes and not imply the relative importance of associations in terms of p-values (i.e., significance tests).

Response: Thank you for this suggestion. We agree that this is a good idea, and we have now updated the introduction to detail the size of the association of the studies we refer to. We have added additional detail to the degree of attenuation seen in the Möttus et al.’s (2012, *Health Psychol*, 33:164-73) study examining the role of cognitive ability in the relationship between health literacy and physical health. We also report the size of the risk of death for individuals with inadequate health literacy in models with and without cognitive function in Baker et al. (2008, *J Gen Intern Med*, 23:723-6).

Page 5: *“Addition of cognitive ability in older age significantly attenuated the association between functional health literacy with physical fitness by 43%, and number of natural teeth by 39%; however, it did not attenuate the association between functional health literacy and body mass index (BMI).”*

Page 5: *“In the fully adjusted model which included childhood and late adulthood cognitive ability, as well as other early-life factors, the association between functional health literacy and physical fitness, though attenuated by 43%, remained significant,[25] suggesting that functional health literacy may play a small but unique role in physical fitness.”*

Page 6: *“When not controlling for cognitive function, Baker et al.[32] found that individuals with inadequate compared to adequate health literacy had a 50% higher risk of dying. When additionally adjusting for cognitive function, the risk reduced to 27%, but remained significant.”*

3. The three measures of HL used in this study are all unidimensional – focused on various aspects of functional health literacy. The proposed ‘general health literacy measure’ is therefore a composite of functional health literacy. There is a considerable literature that questions the robustness of the REALM even in the country from which it was developed (USA). Purpose built multi-dimensional health literacy tools do exist, are applied widely in UK/EU, and cover 3 to 9 dimensions, e.g., the HLS-EU and the HLQ, respectively.

Response: The composite measure of health literacy is now referred to as “general functional health literacy” throughout to reflect this point.

We have added a paragraph to the discussion detailing the limitations of the functional health literacy tests used here, including the fact some have questioned the robustness of the REALM.

Page 22: “...the NVS has been found to have poor concurrent validity with the Test of Functional Health Literacy in Adults.[39] In support of this, we found that the rank-order correlation between the NVS and S-TOFHLA was modest ($r = 0.44$). Concerns have been raised about the fact that the REALM assesses only the ability to read and pronounce words.[38] Knowing how to pronounce medical words may not be directly related to the ability to understand medical information, and therefore this may not adequately cover all the domains of functional health literacy.[38]”

We have added information about the availability of multi-dimensional health literacy measures.

Page 22: “Indeed, all the tests used here were designed to largely measure the component of health literacy known as functional health literacy. None of these measures assess other components of health literacy such as the skills required to critically analyse health information or the communicative skills needed to participate and navigate in the health-care environment.[3] Assessments of health literacy that are designed to measure a much broader range of health literacy skills are available. The European Health Literacy Survey Questionnaire measures self-reported skills in being able to access, understand, appraise, and apply health-related information in the health-care setting, as well as in disease prevention and health promotion.[46]”

4. The TOFHLA is a timed reading comprehension tests that use the modified Cloze procedure, in which every 5th to 7th word in a passage is omitted and replaced with a blank space. The respondent must select a word to fit into the blank spaces from the 4 multiple-choice options provided for each space. Given that it is a timed test, and requires substantial cognitive capacity, I think it is important that the reader is provided with this information in this setting. The nature of the test, the cognitive challenge, is likely to be a reason for its association with fluid ability. The same goes for the NVS, but to a smaller extent.

Response: We have included information about the fact that the S-TOFHLA is a time test in the methods section.

Page 8: “Participants had 12 minutes to complete both sections.”

We have also updated the discussion to highlight to readers that the S-TOFHLA and NVS are cognitively demanding tasks and a possible reason for the attenuation seen in the relationship between health literacy and mortality seen when fluid cognitive ability is added in the models may be because there is likely to be an overlap in the content of the health literacy and fluid ability tests.

Page 21-22: “This attenuation is likely to be because there is an overlap in the content of tests of fluid ability and the NVS and S-TOFHLA. The NVS and S-TOFHLA are cognitively demanding tasks that are likely to be substantially measuring fluid-type cognitive abilities, such as working memory and reasoning, that decline with increasing age.[15]”

5. Given that education is a well-known correlate of health literacy (a potential confounder) I do not understand why this isn’t included in model 2 onward. This needs to be part of the basic model.

Response: We did not originally include this in the basic models as we thought that childhood intelligence would capture the early life ability, which quite strongly predicts how long people spend in education and the level of their attained qualifications. However, we have now updated the

models so that education is included in model 2, after adjusting for age and sex, and before adjusting for age 11 IQ. Adding education in model 2 did not change the pattern of associations found here.

Reviewer 2 comment: Reporting correlations to 3 decimal places seems like spurious accuracy.

Response: We have edited Table 3 and Supplementary Table 1 to report correlations to 2 decimal places.

6. The inclusion of the so called General health literacy, i.e., the composite of the 3 measures, and little to the granularity or robustness of the data analysis.

Response: We have included the general (functional) health literacy measure because we wanted a composite measure of health literacy that captures the shared variance between the tests. Whereas all the tests are used here to measure functional health literacy, they all involve different tasks. The REALM assesses the ability to read and pronounce words, whereas the S-TOFHLA and NVS assess reading comprehension and numeracy in a health context. These are different-seeming tasks, but there is substantial shared variance between these three tests (59.7%); therefore we judge that this is a good measure of general functional health literacy.

We have updated the methods section to make it clear to the read that we are using this measure to capture the shared variance between the three functional health literacy tests.

Page 9: *“The three functional health literacy measures used here have been found to correlate moderately with each other.[25] To capture the shared variance between these tests, a general measure of functional health literacy was created by entering scores on the three tests into a principal component analysis (PCA).”*

Page 9: *“This component accounted for 59.7% of the total variance, confirming there was substantial shared variance between the three functional health literacy tests.”*

7. Remove the redundant 1.00 in table 3.

Response: These have been removed from Table 3 and Supplementary Table 1.

8. A simple content analysis and structural analysis of the HL and cognitive tests used seems prudent. Rather than the content of the HL measures, it may be the kinds of tasks required to complete the tests leads to the strong association between HL and fluid ability.

Response: We agree that it might be the kinds of tasks required to complete the tests that may have led to the strong association between HL and fluid ability. We have updated the discussion to reflect that there is likely some overlap in the content of the measures of HL and the measures of fluid ability and this is likely to explain association between these tests.

Pages 21-22: *“This attenuation is likely to be because there is an overlap in the content of tests of fluid ability and the NVS and S-TOFHLA. The NVS and S-TOFHLA are cognitively demanding tasks that are likely to be substantially measuring fluid-type cognitive abilities, such as working memory and reasoning, that decline with increasing age.[15]”*

9. The discussion should outline the limitations of the HL tools used, especially the coverage of one of the three broad areas of HL, i.e., functional health literacy, rather than the many other domains that represent the full construct.

Response: We added a paragraph to the discussion detailing some of the limitations of the health literacy tools. We highlight that some authors have suggested that the REALM should not be used because it does not adequately measure all areas of functional health literacy.

Page 22: “Some researchers have questioned the validity of some of the functional health literacy tests used here. The Test of Functional Health Literacy in Adults is often reported as the gold standard functional health literacy test.[39] However, the NVS has been found to have poor concurrent validity with the Test of Functional Health Literacy in Adults.[39] In support of this, we found that the rank-order correlation between the NVS and S-TOFHLA was modest ($r = 0.44$). Concerns have been raised about the fact that the REALM assesses only the ability to read and pronounce words.[38] Knowing how to pronounce medical words may not be directly related to the ability to understand medical information, and therefore this may not adequately cover all the domains of functional health literacy.[38]”

We also now highlight to readers that the measures used in our study only assess functional health literacy. We make readers aware that other, more detailed, measures of health literacy are available, such as the European Health Literacy Survey Questionnaire

Page 22: “Indeed, all the tests used here were designed to largely measure the component of health literacy known as functional health literacy. None of these measures assess other components of health literacy such as the skills required to critically analyse health information or the communicative skills needed to participate and navigate in the health-care environment.[3] Assessments of health literacy that are designed to measure a much broader range of health literacy skills are available. The European Health Literacy Survey Questionnaire measures self-reported skills in being able to access, understand, appraise, and apply health-related information in the health-care setting, as well as in disease prevention and health promotion.[46]”

VERSION 2 – REVIEW

REVIEWER	Richard Osborne Deakin University, Australia
REVIEW RETURNED	27-Jun-2018

GENERAL COMMENTS	The paper is much improved. Only some minor improvements remain.  1. Page 3 line 5 Please be precise. Rather than ‘multiple’, state ‘3 functional HL tests’. The authors state that the measures assess ‘...different aspects of functional HL’. The specific components of functional HL are not well defined. There is no/limited research on the specific components of functional HL – perhaps it is fully covered by the TOFHLA. Therefore it is best to say ‘different measures’ of functional HL rather than ‘different components’. The same components might being measured by the 3 tests used, with different precision. 2. P3116 insert ‘functional’ before ‘health literacy’ 3. P3127 given the argument p417 “This overlap is so strong that some have proposed that functional health literacy should not be considered a unique construct but, instead, should be thought of as a specific component of cognitive function” it would seem prudent to highlight ‘whether’ functional health literacy is independent of cognitive function. In the Methods the authors further state the measures are highly correlated $r \sim 0.8$, i.e., measuring the same construct / same component. 4. P19133 the authors suggest 4 measures of functional HL are used. It is more accurate to say that 3 were used, and a composite of the 3. This will then be consistent with p21116. P21127 is then not consistent. 5. P22141 in practice, the European questionnaire has limited validity testing and includes only 3 highly correlated scales, and was designed to measure country-level differences with limited comprehensive psychometric testing. More robust tools, used more
--

	widely, exist that cover distinct and much broader indicators of health literacy, namely the HLQ.
--	---

VERSION 2 – AUTHOR RESPONSE

Review 2

1. Page 3 line 5 Please be precise. Rather than ‘multiple’, state ‘3 functional HL tests’. The authors state that the measures assess ‘...different aspects of functional HL’. The specific components of functional HL are not well defined. There is no/limited research on the specific components of functional HL – perhaps it is fully covered by the TOFHLA. Therefore it is best to say ‘different measures’ of functional HL rather than ‘different components’. The same components might being measured by the 3 tests used, with different precision.

Response: We thank Reviewer 2 for their additional comments. We have now updated the “Strengths and limitations of this study” section to state that we used “three functional health literacy tests”. We have also removed the word “component”.

Page 3: *“This study used three functional health literacy tests, which enabled us to create a composite functional health literacy measure.”*

2. P3116 insert ‘functional’ before ‘health literacy’

Response: We have added “functional” to this sentence.

Page 3: *“This study had comprehensive tests of cognitive ability measured in both childhood and old age which allowed us to investigate whether childhood and old age cognitive ability independently played a role in the relationship between functional health literacy and mortality.”*

3. P3127 given the argument p417 “This overlap is so strong that some have proposed that functional health literacy should not be considered a unique construct but, instead, should be thought of as a specific component of cognitive function” it would seem prudent to highlight ‘whether’ functional health literacy is independent of cognitive function. In the Methods the authors further state the measures are highly correlated $r \sim 0.8$, i.e., measuring the same construct / same component.

Response: Whereas the original paper describing the S-TOFHLA found a correlation of 0.8 between this test and the REALM, in our study the three health literacy tests (REALM, S-TOFHLA, NVS) correlated only moderately with each other ($r = 0.35$ to 0.44 ; see Table 3). These same tests show higher correlations with cognitive ability than with each other. The correlation between the REALM and age 11 IQ is $r = 0.44$, and the correlation between the S-TOFHLA and the NVS with current fluid cognitive ability is $r = 0.55$.

Following journal guidelines, we have not added whether functional health literacy is independent of cognitive function to the section “Strengths and limitations of this study” as suggested above because a) we do think that our finding that health literacy and cognitive ability are strongly associated is either a strength or limitation, and b) the BMJ Open author guidelines state that this section “should not include the results of the study.”

We have, however, responded to the comment’s suggestion by expanding the discussion section to further highlight to the reader that our results suggest that health literacy is mostly not independent of cognitive function. We have also updated this section to highlight that while the original S-TOFHLA paper found a high correlation between the S-TOFHLA and the REALM, more moderate correlations between these tests have been reported elsewhere.

Pages 21-22: *“The results of this study support the proposal by Reeve and Basalik[26] that functional health literacy may not be a unique construct; instead, it is tenable that tests of functional health literacy may in fact be measuring cognitive ability. First we found, as has been*

reported elsewhere,[23, 25] that tests of health literacy tended to correlate more strongly with tests of cognitive ability than with each other. The original paper describing the S-TOFHLA found that this test correlated with the REALM at $r = 0.80$,[6] suggesting these tests are measuring the same underlying ability. However, other studies have found moderate correlations between these tests, similar to ours.[46] Second, we found that the NVS, S-TOFHLA and general functional health literacy no longer predicted mortality when accounting for fluid ability. The results of our study suggest that health literacy may not be independent of cognitive function. This attenuation is likely to be because there is an overlap in the content of tests of fluid ability and the NVS and S-TOFHLA. The NVS and S-TOFHLA are cognitively demanding tasks that are likely to be substantially measuring fluid-type cognitive abilities, such as working memory and reasoning, that decline with increasing age.[15] Childhood cognitive ability did not attenuate the association between functional health literacy, suggesting that the NVS and S-TOFHLA are measuring current fluid-type cognitive capability in old age, and not lifelong intelligence. Current fluid ability in old age may be driving the association between functional health literacy and mortality simply because tests of functional health literacy are assessing the same underlying abilities as measures of fluid ability.”

4. P19I33 the authors suggest 4 measures of functional HL are used. It is more accurate to say that 3 were used, and a composite of the 3. This will then be consistent with p21I16. P21I27 is then not consistent.

Response: We have now updated the discussion section of the manuscript to state that three measures of functional health literacy were used in this study and we used these three measures to create a composite functional health literacy measure.

Page 19: “Three measures of functional health literacy were used; the REALM, S-TOFHLA, and NVS. These three measures were also used to create a composite measure of functional health literacy.”

5. P22I41 in practice, the European questionnaire has limited validity testing and includes only 3 highly correlated scales, and was designed to measure country-level differences with limited comprehensive psychometric testing. More robust tools, used more widely, exist that cover distinct and much broader indicators of health literacy, namely the HLQ.

Response: We now detail both the ELS-EU-Q and the HLQ as other possible health literacy measures that assess a much broader range of health literacy skills.

Page 22: “Assessments of health literacy that are designed to measure a much broader range of health literacy skills are available, such as the Health Literacy Questionnaire (HLQ)[47] and the European Health Literacy Survey Questionnaire (HLS-EU-Q).[48] The HLQ assesses nine dimensions of health literacy, including the ability to actively manage health and navigate the health care system.[47] Whereas the HLS-EU-Q measures self-reported skills in being able to access, understand, appraise, and apply health-related information in the health-care setting, as well as in disease prevention and health promotion.[48] Fluid cognitive ability may not play a role in the association between health literacy and mortality if we used these self-reported, broad, measures of health literacy, rather than the objective, but narrow, functional health literacy tests used here.”

VERSION 3 – REVIEW

REVIEWER	Richard Osborne Deakin University, Australia
REVIEW RETURNED	18-Jul-2018
GENERAL COMMENTS	The authors have satisfactorily responded to my suggestions.